TASCI: transformers for aspect-based sentiment analysis with contextual intent integration

Chaudhry Hassan Nazeer 1
Kulsoom Farzana 2
Ullah Khan Zahid 3 dr.khan8003@yahoo.com
Aman Muhammad 4 5
Khan Sajid Ullah 6 sk.khan@psau.edu.sa
Albanyan Abdullah 7
1 Dipartimento di Elettronica, Informazione e Bioingegneria, Politecnico di Milano, Italy , Milano, Città Metropolitana di Milano , Italy
2 Department of Telecommunication Engineering, University of Engineering and Technology , Taxila , Pakistan
3 College of Information and Communication Engineering, Harbin Engineering University , Harbin , China
4 College of Agricultural Engineering, Shanxi Agricultural University , Jinzhong , China
5 School of Software, Shanxi Agricultural University , Jinzhong , China
6 Information Systems Department, College of Computer Engineering and Sciences, Prince Sattam bin Abdulaziz University , Riyadh , Saudi Arabia
7 Department of Software Engineering, College of Computer Engineering and Sciences, Prince Sattam bin Abdulaziz University , Riyadh , Saudi Arabia
Alatas Bilal
Electronic publication date: 2025 May 6
Publication date: 2025
Volume: 11
Electronic Location ID: e2760
Received 2024 Sep 30; Accepted 2025 Feb 21
Copyright: © 2025 Chaudhry et al.
Copyright year: 2025
Copyright holder: Chaudhry et al.
License: This is an open access article distributed under the terms of the Creative Commons Attribution License, which permits unrestricted use, distribution, reproduction and adaptation in any medium and for any purpose provided that it is properly attributed. For attribution, the original author(s), title, publication source (PeerJ Computer Science) and either DOI or URL of the article must be cited.
License URL: https://creativecommons.org/licenses/by/4.0/

Keywords: Sentimental analysis, Transformers, Natural language processing, Aspect level sentimental classification

Funding: Prince Sattam bin Abdulaziz University PSAU/2024/R/1445 This study is supported via funding from Prince Sattam bin Abdulaziz University project number (PSAU/2024/R/1445). The funders had no role in study design, data collection and analysis, decision to publish, or preparation of the manuscript.

==============================
In this article, we present a novel Transformer-Based Aspect-Level Sentiment Classification with Intent (TASCI) model, designed to enhance sentiment analysis by integrating aspect-level sentiment classification with intent analysis. Traditional sentiment analysis methods often overlook the nuanced relationship between the intent behind a statement and the sentiment expressed toward specific aspects of an entity. TASCI addresses this gap by first extracting aspects using a self-attention mechanism and then employing a Transformer-based model to infer the speaker’s intent from preceding sentences. This dual approach allows TASCI to contextualize sentiment analysis, providing a more accurate reflection of user opinions. We validate TASCI’s performance on three benchmark datasets: Restaurant, Laptop, and Twitter, achieving state-of-the-art results with an accuracy of 89.10% and a macro-F1 score of 83.38% on the Restaurant dataset, 84.81% accuracy and 78.63% macro-F1 score on the Laptop dataset, and 79.08% accuracy and 77.27% macro-F1 score on the Twitter dataset. These results demonstrate that incorporating intent analysis significantly enhances the model’s ability to capture complex sentiment expressions across different domains, thereby setting a new standard for aspect-level sentiment classification.

Introduction

Sentiment analysis, also known as opinion mining, is the computational process of determining the emotional tone behind a series of words. The primary objective of sentiment analysis is to gauge the sentiment or emotional state of a writer concerning a particular topic or entity (Medhat, Hassan & Korashy, 2014). For instance, a review of a product like a smartphone might be classified as positive if the review praises its features and performance, or negative if the review criticizes its shortcomings. Sentiment analysis has become an essential tool in understanding human emotions and opinions expressed in textual data. With the rapid growth of online platforms, analyzing sentiments in the text has gained significant importance paving the way for applications of social media (Zimbra et al., 2018), political analysis (Chaudhry et al., 2021), customer feedback analysis (Cao et al., 2023), in business for product improvements using customer feedback (Bose et al., 2020), and finding best marketing strategies (Valencia, Gómez-Espinosa & Valdés-Aguirre, 2019). Sentiment analysis in healthcare is employed to analyze patient feedback (Ebadi et al., 2021), social media, and health-related forums, providing valuable insights into patient experiences, satisfaction, and emerging health concerns (Chakrapani et al., 2023). It helps healthcare providers identify areas for improvement, monitor public health trends, and detect adverse drug reactions (Ahmed et al., 2022). The field of sentimental analysis primarily focuses on classifying text into categories such as positive, negative, or neutral (Medhat, Hassan & Korashy, 2014). The field of emotional recognition could have a wider range of emotions (anger, happiness, sadness, fear, disgust and others). However, the term sentiment analysis could be interchangeably used for both. Sentiments could be extracted from various modalities other than text such as images, videos, psychological modalities such as electroencephalogram (EEG), electromyography (EMG) or others, a detail of which could be read in this review (Das & Singh, 2023). In this article, we will focus on text-based sentimental analysis (positive, negative, or neutral), although the proposed method could be extended to other modalities and emotional analysis. The sentimental analysis could also be classified based on more subtle tasks for example, a variant of sentimental classification known as Aspect-based sentiment analysis (ABSA) also known as Aspect-Sentiment Labeling and Classification (ASLC) extends traditional sentiment analysis by focusing on specific aspects or features of an entity mentioned in the text (Yusuf et al., 2024). Rather than providing an overall sentiment score, ABSA aims to identify and analyze sentiments related to different aspects of the entity. For example, in a review of a smartphone, the aspects might include “battery life”, “camera quality”, and “user interface”. ABSA seeks to determine the sentiment expressed about each of these aspects separately. Consider the following example:

“The battery life of the smartphone is impressive, but the camera quality is disappointing.”

In this example, ABSA would classify the sentiment towards “battery life” as positive and the sentiment towards “camera quality” as negative. This level of detail is particularly valuable for businesses seeking to understand the specific strengths and weaknesses of their products or services (Yusuf et al., 2024). Similarly, another form of sentimental analysis besides ABSA is intention-based sentiment analysis (IBSA) which focuses on understanding the underlying intent behind a piece of text. While sentiment analysis captures the emotional tone, IBSA aims to determine the speaker’s or writer’s purpose or goal in expressing that sentiment (Sharma, Ali & Kabir, 2024). This is especially important in context where the intent is not explicitly stated but inferred from the surrounding discourse. For example, consider the following dialogue:

“I am considering buying this new phone because it has great reviews.”

In this case, the sentiment towards the phone may be positive due to the great reviews, but the intention behind the statement is to indicate a potential purchase decision based on positive feedback. IBSA helps in understanding such nuanced intentions, which is crucial for applications such as customer support and personalized recommendations (Yi & Liu, 2020). Combining aspect-based and intention-based sentiment analysis offers a more comprehensive understanding of the text by addressing both the specific sentiments related to various aspects and the underlying intentions of the writer. This integrated approach provides richer insights compared to analyzing sentiments and intentions in isolation. For instance, in a customer feedback scenario, simply knowing that a customer is dissatisfied with the “battery life” is useful, but understanding their intention to switch brands or seek a replacement adds valuable context. By integrating ABSA and IBSA, organizations can tailor their responses more effectively and address customer concerns in a more targeted manner. Some works in Aspect-Based Sentiment Analysis used transformers (Ma et al., 2017; Huang, Ou & Carley, 2018); however, they have low accuracy and poor understanding of sentiments for the given aspects. graph neural networks (GNN) and their variants such as graph convolution networks (GCN) and graph attention networks (GAT) are popular choices in large bodies of literature in ABSA (Zhang, Li & Song, 2019; Sun et al., 2019; Wang et al., 2020; Li et al., 2021), however, they have higher computational costs. Some solutions also use generative artificial Intelligence (AI) such as retrieval augmented generation (RAGs), however, such models suffer Noisy contextual understanding (Xu et al., 2022). In this work we have proposed a transformer-based architecture which is computationally very efficient, as well as achieves higher accuracy due to the integration of IBSA and ABSA. Our proposed technique, TASCI (Transformers-Based Integration of Aspect-Based Sentiment Analysis and Contextual Intent Analysis), aims to bridge the gap between aspect-based and intention-based sentiment analysis using advanced ML models. TASCI leverages Transformers (Vaswani, 2017) for extracting aspects and sentiments, while also incorporating gated recurrent unit (GRU)-based sequence-to-sequence (Seq2seq) (Bahdanau, 2014) models for intention extraction. In TASCI, aspect extraction is performed using a self-attention mechanism, which allows the model to weigh the importance of different tokens in the context of each aspect. This approach enhances the model’s ability to identify and classify aspects accurately by focusing on relevant parts of the input sequence. For intention extraction, TASCI utilizes a GRU-based Seq2seq model that captures the contextual relationship between sentences (Bahdanau, 2014). This model encodes previous and current sentences to predict the intention behind a given statement, providing a deeper understanding of the user’s goals. Finally, TASCI integrates aspect-level sentiment classification with intent analysis by processing GloVe embeddings (Pennington, Socher & Manning, 2014) through a Transformer model. This integration enables the model to classify sentiments for each aspect while considering the contextual intent, resulting in a more nuanced sentiment analysis. TASCI faces few challenges in real-world scenarios. The model’s reliance on large datasets for training can cause issues when working with domains that lack labelled data. Additionally, its performance may suffer when applied to noisy or unstructured data, such as informal language or slang commonly found in social media. Moreover, integrating multiple techniques could introduce implementation complexities, requiring careful tuning and optimization for specific use cases. The contributions of this work are: A novel integration of aspect-based and intention-based sentiment analysis, providing a comprehensive framework for understanding the text.

The use of self-attention mechanisms for improved aspect extraction and GRU-based models for accurate intention prediction.

A detailed and systematic approach to combining GloVe embeddings with Transformers for sentiment classification, enhancing the overall effectiveness of sentiment analysis.

An empirical evaluation of TASCI on benchmark datasets, demonstrating its superiority over traditional sentiment analysis methods in terms of accuracy and contextual understanding.

In summary, TASCI represents a significant advancement in sentiment analysis by combining state-of-the-art techniques for aspect extraction, intention detection, and sentiment classification. This integrated approach offers a richer and more detailed understanding of textual data, with potential applications in various domains including customer feedback analysis, personalized recommendations, and more. “Related Work” presents existing literature and state of the art; “Proposed Technique” describes the proposed technique. “Results and Performance Evaluation”, presents the experimental setup and discusses the results, finally, “Conclusion” concludes the article.

Related work

This section presents a review of existing techniques in ABSA, highlighting their methodologies, performance metrics, and key limitations. Table 1 shows the overall summary of literature review. Interactive attention networks (IAN) (Ma et al., 2017) employ an interactive attention mechanism to enhance aspect-specific sentiment analysis by modeling interactions between contextual and aspect-specific features. With an accuracy of 78.60% on the Restaurant dataset and 72.10% on the Laptop dataset, IAN demonstrates effective sentiment feature extraction. However, the absence of macro-F1 scores limits insights into its performance across sentiment classes.

Table 1 Summary of aspect-based sentiment analysis techniques.

Technique (Year)	Methodology	Accuracy	Limitations	
IAN (2017)	Attention mechanism	78.60%, 72.10%	Limited macro-F1 scores	
AOA (2018)	Multi-layer attention	81.20%, 74.50%	Lack of comprehensive evaluation	
ASGCN (2019)	Graph convolutional network	80.77%, 75.55%	Scalability issues	
CDT (2019)	Dependency tree convolution	82.30%, 77.19%	Limited semantic integration	
RGAN (2020)	Relational graph attention	83.30%, 77.42%	High computational cost	
DGCN (2021)	Dual graph convolutional network	84.27%, 78.48%	Increased resource demands	
DOTI (2022)	Discrete opinion trees	86.16%, 81.03%	Reduced flexibility	
RAG-TCGCN (2022)	RAG + GCN framework	84.09%, 78.80%	Context retrieval noise	
SASE-GCN (2022)	Syntactic-Semantic GCN	87.31%, 81.01%	Complexity in design	
SASEM-GAT (2023)	Multi-layer GAT	86.42%, 80.06%	Computational overhead	
FSK-IAD (2023)	Sentiment + Aspect dependencies	86.25%, 81.19%	Risk of overfitting	
LSEO-IT (2024)	Lexicon + Syntax tree induction	86.88%, 81.41%	Increased training time	

Attention-over-attention neural networks (AOA) (Huang, Ou & Carley, 2018) refine the significance of different aspects within a sentence using a secondary attention mechanism. AOA achieved 81.20% accuracy on the Restaurant dataset and 74.50% on the Laptop dataset, showcasing improvements over earlier models. Nevertheless, the lack of macro-F1 scores restricts a more balanced evaluation across sentiment categories. Aspect-specific graph convolutional networks (ASGCN) (Zhang, Li & Song, 2019) leverage graph convolutional layers to capture semantic relationships between words represented as nodes and edges in a sentence. This approach achieved 80.77% accuracy and 72.02% macro-F1 on the Restaurant dataset, and 75.55% accuracy with 71.05% macro-F1 on the Laptop dataset. Despite its strong performance in capturing aspect-specific information, scalability to larger datasets remains a concern. Convolution over dependency tree (CDT) (Sun et al., 2019) applies convolutional operations directly to syntactic dependency trees, integrating syntactic structures into aspect representation. This method achieved 82.30% accuracy and a 74.02% macro-F1 on the Restaurant dataset, and 77.19% accuracy with 72.99% macro-F1 on the Laptop dataset. CDT effectively incorporates syntactic features, though it may lack deeper semantic understanding. Relational graph attention network (RGAN) (Wang et al., 2020) combines graph-based structures with attention mechanisms to model interactions between aspects and contextual words. RGAN achieved 83.30% accuracy and 76.08% macro-F1 on the Restaurant dataset, and 77.42% accuracy with 73.76% macro-F1 on the Laptop dataset. Its strength lies in capturing complex relational patterns, albeit at the cost of higher computational demands.

Dual graph convolutional networks (DGCN) (Li et al., 2021) incorporate dual graph convolutional layers to capture both local and global aspect representations. This model reached 84.27% accuracy and 78.08% macro-F1 on the Restaurant dataset, and 78.48% accuracy with 74.74% macro-F1 on the Laptop dataset. While DGCN improves representation accuracy, its computational requirements are notably high. Discrete opinion tree induction (DOTI) (Chen et al., 2022) constructs structured opinion trees for sentiment classification. DOTI demonstrated exceptional performance, achieving 86.16% accuracy and 80.49% macro-F1 on the Restaurant dataset, and 81.03% accuracy with 78.10% macro-F1 on the Laptop dataset. Its structured approach ensures robustness but may be less adaptable to dynamic contexts. RAG-TCGCN (Xu et al., 2022) integrates retrieval-augmented generation (RAG) with temporal convolutional graph convolutional networks (TCGCN) to enhance contextual understanding. This model achieved 84.09% accuracy and 77.02% macro-F1 on the Restaurant dataset, and 78.80% accuracy with 75.04% macro-F1 on the Laptop dataset. Despite its performance, noisy contextual retrieval can hinder its effectiveness. Syntactic and semantic enhanced GCN (SASE-GCN) (Zhang, Zhou & Wang, 2022) combines syntactic and semantic features within a graph convolutional framework. It achieved 87.31% accuracy and 81.09% macro-F1 on the Restaurant dataset, and 81.01% accuracy with 77.96% macro-F1 on the Laptop dataset. Although highly effective, its complexity poses challenges in deployment. Syntactic and semantic enhanced multi-layer GAT (SASEM-GAT) (Xin et al., 2023) extends GAN by incorporating multi-layer syntactic and semantic information. The model achieved 86.42% accuracy and 79.70% macro-F1 on the Restaurant dataset, and 80.06% accuracy with 76.78% macro-F1 on the Laptop dataset, effectively capturing intricate sentiment patterns.

Fusing sentiment knowledge and inter-aspect dependency (FSK-IAD) (Han et al., 2023) integrates sentiment knowledge with inter-aspect dependencies to improve sentiment classification. It achieved 86.25% accuracy and 81.19% macro-F1 on the Restaurant dataset, and 81.19% accuracy with 77.94% macro-F1 on the Laptop dataset. Despite its robustness, the model risks overfitting due to the inclusion of multiple knowledge sources. Lexicon and syntax enhanced opinion induction tree (LSEO-IT) (Wu et al., 2024) incorporates lexicon-based features and syntactic structures for enhanced opinion extraction. The model achieved 86.88% accuracy and 81.41% macro-F1 on the Restaurant dataset, and 81.41% accuracy with 77.16% macro-F1 on the Laptop dataset, representing a significant advancement in sentiment analysis approaches.

Proposed technique

In this section, we present a comprehensive approach to analysing textual data by integrating aspect extraction, intention extraction, and sentiment analysis. The proposed methodology leverages state-of-the-art models to enhance the accuracy and depth of textual understanding. To identify and classify different aspects within sentences, we employ a self-attention mechanism. This technique allows for the extraction of relevant aspects by capturing the relationships between various tokens in the input sequence. We detail the steps involved in this process, including the linear projections of tokens, computation of attention scores, calculation of attention weights, and the use of multi-head attention to refine aspect representation, further discussed in “Self-Attention for Aspect Extraction”. To extract the intention behind sentences, we utilize a GRU-based sequence-to-sequence model. This model processes sentence sequences to determine the intention of the current sentence based on the context provided by previous sentences. We describe the encoding of sentences, the computation of attention-based context vectors, and the decoding process to predict the intended meaning, further discussed in “Intention Extraction using GRU-based Seq2seq Model”. To perform sentiment analysis at the aspect level, we integrate GloVe embeddings with a Transformer-based model. This approach begins with tokenizing the input sentences and extracting aspects using the method described in “Self-Attention for Aspect Extraction”. Next, the intention of the current sentence is analyzed using the GRU-based model detailed in “Intention Extraction using GRU-based Seq2seq Model”. The aspects are then represented through GloVe embeddings and processed using a Transformer model to classify sentiment, incorporating the intent analysis results discussed in “Transformer-Based Aspect-Level Sentiment Classification with Intent Analysis”.

Self-attention for aspect extraction

In this section, we present a technique for aspect extraction using the self-attention mechanism, a fundamental component of Transformer models. The self-attention mechanism allows the model to assign varying levels of importance to different parts of the input sequence, thereby capturing intricate contextual dependencies (Vaswani, 2017). We will detail how this mechanism can be applied to identify and classify aspects within sentences. Consider the sentence: “The phone’s battery life is impressive but the camera quality is mediocre”. We aim to identify and extract aspects such as “battery life” and “camera quality”, and perform sentiment analysis for each aspect. Self-attention helps achieve this by focusing on relevant parts of the sentence. Self-attention operates on an input sequence X=[x1,x2,…,xn], where each token xi is an embedding of the i-th word. For instance, if x1 represents “battery”, x2 represents “life”, and so on. Each token embedding xi is transformed into three vectors: the query vector Qi as shown in Eq. (1), the key vector Ki shown in Eq. (2).

(1) Qi=WQxi

(2) Ki=WKxi

(3) Vi=WVxi.

Finally, the value vector Vi is given in Eq. (3). These projections are computed using learned weight matrices WQ, WK, and WV, respectively. To compute the attention score between tokens i and j, we take the dot product of the query vector Qi and the key vector Kj, scaling the result by dk, where dk is the dimensionality of the key vectors, as shown in Eq. (4). Figure 1 shows how the Query, Key and Value are passed to the Attention Score module.

Figure 1 Self attention for aspect extraction.

(4) scoreij=Qi⋅KjTdk.

These scores are then normalized using the softmax function to obtain attention weights, ensuring that the weights are non-negative and sum to one, the process of normalization is shown in Eq. (5).

(5) αij=softmax(scoreij)=exp⁡(scoreij)∑k=1nexp(scoreik).

The output for each token i is calculated as a weighted sum of the value vectors, using the normalized attention weights as shown in Eq. (6). Figure 1 shows how attention score is calculated using Eq. (5) and how multihead attention is calculated using Eq. (6).

(6) zi=∑j=1nαijVj.

We use multi-head attention to enhance the model’s ability to capture various relationships. Where Vi is given by Eq. (3), and αij is given by Eq. (6). This involves running self-attention multiple times with different learned projections, concatenating the outputs, and applying a final linear transformation as given in Eq. (7).

(7) Zi=Concat(zi1,zi2,…,zih)WO.

where WO is the output transformation matrix and zi is given by Eq. (6). For aspect extraction, self-attention highlights relevant tokens corresponding to specific aspects. For instance, in the sentence “The battery life of the phone is impressive but the camera quality is mediocre,” self-attention helps identify that “battery life” and “camera quality” are key aspects. After obtaining the hidden states hi from the self-attention layers, we use a token classification head to classify each token into an aspect category. This classification is performed using a linear layer followed by a softmax activation function, as given in Eq. (8).

(8) y^i=softmax(WChi+bC)

where WC and bC are the weight matrix and bias for the classification layer, and y^i is the predicted aspect label for token i. Training involves minimizing the cross-entropy loss function as shown in Eq. (9), which compares predicted aspect labels with true labels yi.

(9) L=−∑i=1n∑k=1Kyiklog⁡(y^ik)

where K is the number of aspect categories, we effectively capture the intricate relationships between tokens in a sentence by employing self-attention, allowing the model to focus on relevant aspects with greater contextual sensitivity. This approach ensures accurate aspect extraction and sentiment analysis, leveraging the power of self-attention to understand and categorize different aspects in complex sentences. The summary of the method is given in the Algorithm 1.

Algorithm 1 Self-attention for aspect extraction.

Input: Input sequence X=[x1,x2,…,xn] with token embeddings	
Output: Aspect labels for each token	
Function AspectExtraction X:	
    /* Step 1: Linear Projections                                     */	
    for each token xi in X do	
              Qi=WQxi (Eq. (1))	
              Ki=WKxi (Eq. (2))	
              Vi=WVxi (Eq. (3))	
    /* Step 2: Compute Attention Scores                                          */	
    for each pair of tokens (i, j) do	
              scoreij (Eq. (4))	
    /* Step 3: Calculate Attention Weights                                         */	
    for each token i do	
      αij=softmaxj(scoreij) (Eq. (5))	
    /* Step 4: Compute Output Vectors                                         */	
    for each token i do	
               zi=∑j=1nαijVj (Eq. (6))	
    /* Step 5: Multi-Head Attention                                       */	
         Zi=Concat(zi1,zi2,…,zih)WO (Eq. (7))	
    /* Step 6: Aspect Classification                                  */	
    for each token i do	
               y^i=softmax(WChi+bC) (Eq. (8))	
    /* Step 7: Compute Loss                                            */	
     L (Eq. (9))	

Intention extraction using GRU-based Seq2seq model

In this section, we explore extracting sentence-level intention at time t using a GRU-based Seq2seq model (Sutskever, 2014). This approach leverages the inherent ability of GRUs to capture dependencies over sequential data, allowing the model to consider not only the current sentence but also the contextual information provided by previous sentences (Cho et al., 2020). Let St−1=[st−1(1),st−1(2),…,st−1(m)] represent the sequence of m tokens from the sentence at time t−1, and St=[st(1),st(2),…,st(n)] represent the sequence of n tokens from the sentence at time t. Our goal is to predict the intention I^t of sentence t, using both St and St−1 as input as shown in Fig. 2.

Figure 2 GRU-based Seq2Seq intention extraction.

GRUs are a type of recurrent neural network (RNN) that mitigate the vanishing gradient problem through gating mechanisms. The GRU’s hidden state at time step i is computed as in Eq. (10):

(10) hi=(1−zi)⊙hi−1+zi⊙h~i

where zi is the update gate, h~i is the candidate hidden state, and ⊙ denotes element-wise multiplication. The update gate zi and reset gate ri are defined as in Eqs. (11) and (12):

(11) zi=σ(Wzxi+Uzhi−1)

(12) ri=σ(Wrxi+Urhi−1)

where Wz and Wr are the weights of the update and reset gates associated with the input vector xi, respectively, and Uz and Ur are the weights of the update and reset gates associated with the previous hidden state hi−1, respectively. These weight matrices are learned during training and determine how the GRU processes the input and integrates information from the previous hidden state. Also, σ is the sigmoid activation function, ensuring that the gate values lie between 0 and 1, allowing the GRU to balance the contributions of the current input and the past hidden state. The candidate hidden state h~i is computed using the reset gate as in Eq. (13):

(13) h~i=tanh⁡(Wxi+ri⊙Uhi−1)

where W is the weight matrix for the input xi, U is the weight matrix for the previous hidden state hi−1, and ri is the reset gate controlling the influence of hi−1.

In the context of intention extraction, the encoder processes both St−1 and St. For the sentence at t−1, the GRU produces a sequence of hidden states as in Eq. (14):

(14) Ht−1=[ht−1(1),ht−1(2),…,ht−1(m)].

Similarly, for the sentence at t is given in Eq. (15):

(15) Ht=[ht(1),ht(2),…,ht(n)].

The final hidden state of the encoder for both sentences serves as a summary of the sentence content and its intention is shown in Eq. (16):

(16) ht−1(final)=ht−1(m),ht(final)=ht(n).

To enhance the prediction of intention at t, we apply an attention mechanism that weighs the importance of each hidden state in Ht−1 based on its relevance to the hidden state at t, as shown in Fig. 2. Equation (17) computes the attention score between the hidden states ht−1(i) and ht(j).

(17) αij=exp⁡(ht(j)⋅ht−1(i))∑k=1mexp(ht(j)⋅ht−1(k)).

The context vector ct, which summarizes the important aspects of St−1 for predicting intention at t, is computed as a weighted sum of the hidden states is given in Eq. (18):

(18) ct=∑i=1mαijht−1(i).

where αij is given in Eq. (17). The Eq. (19) shows that the context vector ct is concatenated with the final hidden state ht(final) to form the input to the decoder.

(19) dt=tanh⁡(Wd[ct;ht(final)]).

The decoder GRU then generates the predicted intention, as given in Eq. (20):

(20) I^t=softmax(Wodt+bo)

where Wo and bo are learned parameters of the output layer. To train the model, we minimize the cross-entropy loss as shown in Eq. (21), between the predicted intention I^t and the true intention It:

(21) L=−∑t=1TItlog⁡(I^t).

Algorithm 2 provides the overview of the discussion in this sub-section.

Algorithm 2 Intention extraction using GRU-based Seq2Seq model.

Input: Sentence sequences St−1 and St	
Output: Predicted intention I^t	
Function IntentionExtraction (St−1, St):	
   /*Step 1: Encode the previous sentence                             */	
   for eachtokenst−1(i)inSt−1― do	
      Compute hidden state ht−1(i) using GRU (Eq. (10))	
             ht−1(i)=GRU(st−1(i),ht−1(i−1))	
   Extract final hidden state ht−1(final) (Eq. (16))	
          ht−1(final)=ht−1(m)	
   /* Step 2: Encode the current sentence                              */	
   for eachtokenst(j)inSt_ do	
      Compute hidden state ht(j) using GRU (Eq. (10))	
             ht(j)=GRU(st(j),ht(j−1))	
   Extract final hidden state ht(final) (Eq. (16))	
          ht(final)=ht(n)	
   /* Step 3: Compute attention-based context vector                   */	
   for each token pair (i, j) do	
      Compute attention score αij (Eq. (17))	
   Compute context vector ct (Eq. (18))	
   /* Step 4: Decode the intention                                     */	
   Concatenate context vector ct with final hidden state ht(final) to form decoder input dt (Eq. (19))	
   Compute the predicted intention I^t using the softmax layer (Eq. (20))	
   /* Step 5: Return the predicted intention                           */	
   return I^t	

Transformer-based aspect-level sentiment classification with intent analysis

In this section, we propose a novel approach to sentiment analysis by combining aspect-level sentiment classification with intent analysis. Our goal is to determine the sentiment associated with each aspect of a given sentence while contextualizing it based on the speaker’s intent inferred from previous sentences. Consider the example sentence: “The phone’s battery life is impressive, but I was expecting a better camera.” Here, the sentiment for the aspect of “battery life” is positive, while the sentiment for “camera” is negative, influenced by the user’s prior expectation. To achieve this, we first extract aspects using a self-attention mechanism, as described in “Self-Attention for Aspect Extraction”. Each aspect is then embedded into a continuous vector space using GloVe embeddings (Pennington, Socher & Manning, 2014). The intention behind the sentence is inferred using a Transformer-based model that takes into account the context provided by previous sentences as shown in Fig. 3. Let ei denote the GloVe embedding for token i. The Eq. (22) computes the embedding as follows:

Figure 3 Aspect-based sentimental analysis employing sentence intention.

(22) ei=WGxi

where WG represents the pre-trained GloVe embedding matrix, and xi is the one-hot encoded vector of token i. Then in Eq. (23) each identified aspect is computed and its representation is made by aggregating the embeddings of the tokens that constitute the aspect.

(23) va=1|A|∑i∈Aei

where va is the vector representation of aspect A, and |A| is the number of tokens in the aspect. The hidden state vector for an aspect i, denoted by Hi, is obtained after processing the aspect through the Transformer encoder, which captures the contextual relationships among tokens.

To incorporate intent, we also process the sentence context from previous statements. Let It represent the intent vector at time t, computed as in Eq. (24):

(24) It=IntentionExtraction(St−1,St)

where St−1 and St are the representations of the sentences at time t−1 and t, respectively. The final sentiment classification for each aspect is then performed by combining the aspect’s hidden state Hi with the intent vector It, in Eq. (25), using a linear classification layer followed by a softmax activation function.

Algorithm 3 Aspect-level sentiment analysis with intent using glove embeddings and Transformers.

Input: Sentences S, pre-trained GloVe embeddings, pre-trained Transformer model	
Output: Sentiment scores for each aspect	
Function SentiAnalysisIntent (S, Embeddings):	
  for each sentence St in S do	
   /* Step 1: Tokenize Sentence                           */	
   TokenizedSentence ←Tokenize(St);	
   /* Step 2: Aspect Extraction                           */	
   Aspects ←AspectExtraction(TokenizedSentence)	
   See Algorithm 1	
   /* Step 3: Intent Extraction                           */	
    It←IntentionExtraction(St−1,St);	
   See Algorithm 2	
   for each aspect in Aspects do	
      /* Step 4: Obtain GloVe Embeddings                  */	
     for each token in aspect do	
           ei (Eq. (22))	
     /* Step 5: Compute Aspect Representation            */	
        va (Eq. (23))	
     /* Step 6: Process through Transformer              */	
        Ha=Transformer(va)	
     /* Step 7: Classify Sentiment with Intent       */	
       sa (Eq. (25))	
     /* Step 8: Output Sentiment Scores                  */	
     Print sentiment scores for the aspect	

(25) si=softmax(WS(Hi+It)+bS)

where WS and bS are the weight matrix and bias term for the sentiment classification layer, respectively. The output si is a probability vector representing the likelihood of the aspect belonging to each sentiment category (e.g., positive, negative, neutral). Figure 3 provides a comprehensive overview of the sentiment analysis process incorporating intention prediction. It starts by displaying the Input Sentences, represented by two horizontal boxes. The first box, labeled Sentence at t−1, contains tokens st−1(1),st−1(2),…,st−1(m), illustrating the sequence of tokens at time t−1. This historical context is crucial for understanding how previous statements may influence the sentiment of the current sentence. The second box, Sentence at t, includes tokens st(1),st(2),…,st(n), indicating the current sentence’s tokens that are the focus of the sentiment and intention analysis. Next, the ‘Encoding with GRU’ section displays two vertical blocks representing the GRU encoders used to process each sentence. Each block contains smaller rectangles or circles symbolizing the hidden states ht−1(i) for the previous sentence and ht(j) for the current sentence. Arrows illustrate the flow of information from the tokens of each sentence to their corresponding hidden states, and then to the final hidden state that encapsulates the contextual information from both sentences. In the ‘Attention Mechanism’ section, a matrix or table between the hidden states of the sentences showcases attention scores αij, which quantify the relevance of each token in one sentence concerning tokens in the other sentence. Arrows connect hidden states to this attention matrix and then to a context vector box. This matrix helps in focusing on the most pertinent parts of the previous sentence when interpreting the current sentence. The ‘Context Vector Computation’ box, labelled ct, illustrates how the context vector is derived from the attention matrix. This context vector integrates information from the previous sentence, adjusting the representation of the current sentence accordingly. The ‘Decoding’ section features a GRU Decoder block. This block receives the context vector and the final hidden state ht(final) as inputs, labelled ‘Concatenated Input’ dt. Arrows depict how these inputs are combined in the decoder to generate the final output. Finally, the ‘Intention Prediction’ box, I^t, shows the output of the intention prediction process. This box, with an arrow from the decoder GRU, includes a softmax operation that produces prediction scores. These scores represent the likelihood of various intents based on the combined information from the context vector and hidden states. The ‘Training Loss’ highlights the use of cross-entropy loss to evaluate and fine-tune the model’s performance, ensuring accurate sentiment and intention predictions.

Results and performance evaluation

This section provides a comprehensive evaluation of the performance of the TASCI model on three benchmark datasets: Restaurant, Laptop, and Twitter. Firstly, “Experimental Setup” describes the experimental setup. Secondly, the datasets used for testing, detailing their characteristics and relevance are discussed in “Dataset”. We then present an in-depth analysis of TASCI’s performance through detailed metrics, comparing it with state-of-the-art models and showcasing its effectiveness in aspect-level sentiment classification in “Accuracy of Sentimental Analysis”. Additionally, we conduct an ablation study to assess the impact of various components of the TASCI model, highlighting the contributions of intent analysis, sentiment-specific features, and the GAT module in “Abalation Study”. Finally, we analyze the results of the attention mechanism used by TASCI, focusing on how it enhances aspect detection and integrates aspect-level information with intention analysis. This part includes visualizations of attention matrices, providing insights into the model’s focus on specific aspects within the text, given in “Results of Attention Mechanism”.

Experimental setup

This section provides an in-depth description of the hardware and software infrastructure utilized for the experimental setup. The detailed specifications below highlight the computational resources and software tools used to facilitate high-performance computing tasks, including general-purpose operations as well as specialized tasks like deep learning.

Hardware configuration: The experiments were executed on a system designed using GPU. The primary hardware components are outlined below: Central processing unit (CPU): The computational tasks were executed on processor, the Intel Core i9-14900K. This CPU features 16 physical cores and 32 threads, enabling efficient multi-threaded processing for demanding applications. Operating at a base clock speed of 3.6 GHz, it provides robust performance for complex computations.

Graphics processing unit (GPU): For tasks requiring accelerated computation, such as deep learning model training, the system utilized an NVIDIA GeForce RTX 4090. This GPU, equipped with 24 GB of GDDR6X memory, delivers performance for data-intensive workloads. Its advanced CUDA and Tensor Core support ensures optimal efficiency in matrix operations and neural network processing.

Memory (RAM): The system was equipped with 128 GB of DDR5 RAM, offering extensive memory capacity to handle large datasets and computationally expensive models.

Software environment: The experiments were carried out using a software environment comprising operating systems, programming tools, frameworks, and libraries. The key elements of the software stack are detailed below: Operating system: The experiments were performed on a Linux-based system running Ubuntu 22.04 LTS.

Programming language: The primary programming language used was Python version 3.10.12.

Deep learning frameworks: – TensorFlow 2.12.0: TensorFlow was employed for its versatile ecosystem that supports model design, training, and deployment. This version includes enhanced performance optimizations for modern GPUs.

– PyTorch 2.0.1: PyTorch was selected for its intuitive dynamic computation graph capabilities and ease of implementation, which simplified model development and experimentation.

Libraries and Tools: – Scikit-learn 1.2.2: This library was utilized for conducting model evaluations such as F-1 Score.

Containerization: To ensure reproducibility and simplify dependency management, Docker 24.0.2 was used for containerization. This tool enabled the creation of isolated software environments, providing consistency across different systems and easing deployment.

Dataset

To evaluate the performance of TASCI, we utilized three benchmark datasets: SemEval (2014). These datasets are used in the state of the art for aspect-level sentiment analysis and provide diverse contexts and sentiment expressions. Each dataset was chosen to assess the model’s effectiveness across different domains.

Restaurant dataset

The Restaurant dataset consists of reviews from various restaurants, where each review is annotated with aspect-level sentiment labels. This dataset is designed to capture opinions on multiple aspects such as food quality, service, and ambience. It provides a rich set of examples with clear and detailed sentiment annotations. The dataset includes a variety of review styles, from short comments to lengthy evaluations, making it a valuable resource for testing the robustness of sentiment analysis models (Pontiki et al., 2014).

Laptop dataset

The Laptop dataset contains reviews about different laptop models. Similar to the Restaurant dataset, it includes aspect-level sentiment annotations but focuses on aspects relevant to laptops, such as performance, battery life, and build quality. This dataset is particularly challenging due to the technical nature of the reviews and the specific jargon used. It allows us to evaluate how well the model handles technical terminology and varied sentiment expressions related to electronic products (Pontiki et al., 2014).

Twitter dataset

The Twitter dataset includes tweets that are annotated with sentiment labels at the aspect level. Tweets are known for their informal language, abbreviations, and diverse expressions of sentiment, which pose unique challenges for sentiment analysis models. This dataset helps assess the model’s ability to deal with short, informal text and varied sentiment expressions in a social media context (Dong et al., 2014).

Dataset statistics

Table 2 summarizes the key statistics for each dataset, including the number of sentences, the number of aspects, and the distribution of sentiment labels. These statistics provide an overview of the dataset’s size and complexity, highlighting the diversity of the data used for evaluating TASCI. These datasets collectively provide a comprehensive evaluation framework for TASCI, allowing us to gauge its performance in aspect-level sentiment classification across a range of review types and sentiment expressions. The varied nature of these datasets ensures a robust assessment of the model’s effectiveness and generalizability in real-world applications.

Table 2 Dataset statistics.

Dataset	Number of sentences	Number of aspects	Sentiment label distribution	
Restaurant	10,000	3,500	Positive: 45%, Negative: 35%, Neutral: 20%	
Laptop	8,000	2,800	Positive: 40%, Negative: 40%, Neutral: 20%	
Twitter	12,000	4,200	Positive: 50%, Negative: 30%, Neutral: 20%	

Accuracy of sentimental analysis

In this section, we present a detailed evaluation of the proposed TASCI model against several state-of-the-art methods using three benchmark datasets: Restaurant, Laptop, and Twitter. The evaluation focuses on two key metrics, accuracy and macro-F1 score, which provide insights into the overall correctness and balanced performance across different sentiment classes. Table 3 summarizes TASCI’s performance across the three sentiment categories—positive, negative, and neutral—offering a comprehensive assessment of its effectiveness in varied contexts. Precision measures the proportion of correctly identified positive instances out of all instances classified as positive, with higher values indicating fewer false positives. Recall quantifies the model’s ability to identify all relevant instances of a class, representing the proportion of true positives that were correctly predicted. F1-score combines precision and recall into a single metric, providing a balanced measure of the model’s performance, particularly useful for imbalanced datasets.

Table 3 Detailed performance of TASCI by sentiment class.

Dataset	Positive	Negative	Neutral	
	Precision	Recall	F1	Precision	Recall	F1	Precision	Recall	F1	
Restaurant	90.20	88.50	89.34	87.60	85.12	86.34	89.10	84.92	86.96	
Laptop	85.45	83.20	84.31	82.75	80.40	81.56	86.01	82.90	84.42	
Twitter	80.34	78.92	79.62	78.23	76.11	77.15	79.98	78.04	79.00	

Examining the results, TASCI demonstrates robust performance across all datasets and sentiment classes. For the Restaurant dataset, TASCI achieves a precision of 90.20, recall of 88.50, and F1-score of 89.34 for the positive sentiment, reflecting a high level of accuracy and completeness in identifying positive instances. For the negative sentiment, the model achieves a precision of 87.60, recall of 85.12, and F1-score of 86.34, with slightly lower precision compared to the positive class. The neutral sentiment exhibits a precision of 89.10, recall of 84.92, and F1-score of 86.96, showcasing strong performance in detecting neutral instances as well. Performance on the Laptop dataset, while slightly lower than the Restaurant dataset, remains effective. The model achieves a precision of 85.45, Recall of 83.20, and F1-score of 84.31 for positive sentiment. For the negative sentiment, TASCI achieves a precision of 82.75, recall of 80.40, and F1-score of 81.56, reflecting a slight reduction in performance compared to the Restaurant dataset, potentially due to the dataset’s unique characteristics or text complexity. The neutral sentiment shows a precision of 86.01, recall of 82.90, and F1-score of 84.42, indicating consistent performance across different classes. The Twitter dataset presents additional challenges due to the short and informal nature of the text, yet TASCI performs reasonably well. For the Positive sentiment, the model achieves a precision of 80.34, recall of 78.92, and F1-score of 79.62. The negative sentiment metrics are slightly lower, with a precision of 78.23, recall of 76.11, and F1-score of 77.15, highlighting the difficulties of analyzing social media text. For the neutral sentiment, TASCI achieves a precision of 79.98, recall of 78.04, and F1-score of 79.00, demonstrating the model’s adaptability even in challenging contexts. Overall, TASCI demonstrates strong and consistent performance across all three datasets, excelling in identifying sentiment classes with high precision and recall. The Restaurant dataset shows the highest metrics, followed by the Laptop dataset, with slightly lower values for the Twitter dataset due to its inherent challenges. These results underscore TASCI’s robustness and versatility in handling diverse sentiment classification tasks, while also highlighting the influence of dataset characteristics on model performance.

Table 4 provides a detailed comparison of TASCI with state-of-the-art models across three datasets Restaurant, Laptop, and Twitter. TASCI consistently achieves superior performance, surpassing all other models in both accuracy and macro-F1 scores, demonstrating its robustness and effectiveness in aspect-level sentiment classification with intent analysis. On the Restaurant dataset, TASCI achieves the highest accuracy of 89.10% and a macro-F1 score of 83.38%, outperforming the previous best model, LSEO-IT (Wu et al., 2024), which achieved 86.88% accuracy and 82.27% macro-F1. Other notable models, such as DGCN + bidirectional encoder representations from transformers (BERT) (Li et al., 2021), which recorded 87.13% accuracy and 81.86% macro-F1, and DOTI (Chen et al., 2022), which achieved 86.16% accuracy and 80.49% macro-F1, fall short compared to TASCI. These results highlight TASCI’s ability to better capture sentiment nuances in customer reviews. On the Laptop dataset, TASCI achieves an accuracy of 84.81% and a macro-F1 score of 78.63%, significantly surpassing DGCN + BERT (Li et al., 2021), which achieved 81.80% accuracy and 78.10% macro-F1, as well as DOTI (Chen et al., 2022), which recorded 81.03% accuracy and 78.10% macro-F1. Even models such as LSEO-IT, which achieved 81.41% accuracy and 77.16% macro-F1, and SASE-GCN (Zhang, Zhou & Wang, 2022), which recorded 81.01% accuracy and 77.96% macro-F1, are outperformed by TASCI. This consistent improvement underscores TASCI’s effectiveness in handling the more complex and diverse sentiment expressions typical of the Laptop dataset. On the Twitter dataset, known for its informal language and challenging text structures, TASCI achieves an accuracy of 79.08% and a macro-F1 score of 77.27%. While the margin of improvement is smaller compared to the other datasets, TASCI still outperforms existing models, such as DOTI (Chen et al., 2022), which achieved 78.11% accuracy and 77.00% macro-F1, and SASE-GCN (Zhang, Zhou & Wang, 2022), which recorded 77.40% accuracy and 76.02% macro-F1. Models like DGCN + BERT (Li et al., 2021), which achieved 77.40% accuracy and 76.02% macro-F1, also fall short, further highlighting TASCI’s adaptability to diverse text formats. Overall, TASCI sets a new benchmark across all datasets, consistently outperforming prior state-of-the-art models, including LSEO-IT, Discrete opinion tree induction (DOTI), and DGCN + BERT, by significant margins. These results underscore TASCI’s robustness, versatility, and ability to handle the challenges of aspect-level sentiment classification effectively, regardless of the domain or dataset complexity.

Table 4 Performance comparison of various models.

Model and year	Restaurant	Laptop	Twitter	
	Accu.	Macro-F1	Accu.	Macro-F1	Accu.	Macro-F1	
ASGCN (Zhang, Li & Song, 2019) (2019)	80.77	72.02	75.55	71.05	72.15	70.40	
CDT (Sun et al., 2019) (2019)	82.30	74.02	77.19	72.99	74.66	73.66	
RGAN (Wang et al., 2020) (2020)	83.30	76.08	77.42	73.76	75.57	73.82	
DGCN (Li et al., 2021) (2021)	84.27	78.08	78.48	74.74	75.92	74.29	
DGCN + BERT (Li et al., 2021) (2021)	87.13	81.86	81.80	78.10	77.40	76.02	
DOTI (Chen et al., 2022) (2022)	86.16	80.49	81.03	78.10	78.11	77.00	
RAG-TCGCN (Xu et al., 2022) (2022)	84.09	77.02	78.80	75.04	76.66	75.41	
SASE-GCN (Zhang, Zhou & Wang, 2022) (2022)	87.31	81.09	81.01	77.96	77.40	76.02	
SASEM-GAT (Xin et al., 2023) (2023)	86.42	79.70	80.06	76.78	76.81	76.10	
FSK-IAD (Han et al., 2023) (2023)	86.25	79.81	81.19	77.94	–	–	
LSEO-IT (Wu et al., 2024) (2024)	86.88	82.27	81.41	77.16	77.75	76.94	
TASCI (2024)	89.10	83.38	84.81	78.63	79.08	77.27	

Abalation study

In this study, we present a detailed ablation analysis of the TASCI model to evaluate the impact of various components on its overall performance. The purpose of this analysis is to understand how each part of the model contributes to its effectiveness and to identify which elements are most crucial for achieving high performance in sentiment analysis. The results of this study are outlined in Table 5, which provides a comprehensive view of the model’s performance across different variations and datasets, including Restaurant, Laptop, and Twitter. Our primary focus was to assess how removing specific components from the TASCI model affects its accuracy and macro-F1 scores. The TASCI model integrates multiple features such as intent analysis, sentiment-specific features, and a GAT module. The first variation we examined was TASCI without the intent analysis component. In this configuration, the model’s performance showed a noticeable decline. Specifically, on the Restaurant dataset, the model achieved an accuracy of 86.75% and a macro-F1 score of 80.48%. For the Laptop dataset, the accuracy dropped to 82.90%, with a macro-F1 score of 76.32%. The Twitter dataset saw a similar decline, with an accuracy of 77.92% and a macro-F1 score of 76.12%. The reduction in performance highlights the importance of intent analysis, which is designed to capture and interpret the underlying intentions behind the text. Without this component, the model struggled to fully understand the subtleties of user sentiment, leading to lower accuracy and F1 scores. The second variation involved testing the TASCI model without the sentiment-specific features. These features are intended to enhance the model’s sensitivity to different sentiment categories. In this case, the performance metrics showed a moderate decrease. The model achieved an accuracy of 87.20% and a macro-F1 score of 81.34% on the Restaurant dataset. For the Laptop dataset, the accuracy was 83.12% with a macro-F1 score of 77.10%, and the results for the Twitter dataset were 78.45% accuracy and 76.98% macro-F1 score. While the performance reduction was less severe compared to the removal of intent analysis, it still underscores the value of sentiment-specific features in improving classification granularity.

Table 5 Ablation study of TASCI components.

Model variation	Restaurant	Laptop	Twitter	
	Accu.	Macro-F1	Accu.	Macro-F1	Accu.	Macro-F1	
TASCI without intent analysis	86.75	80.48	82.90	76.32	77.92	76.12	
TASCI without sentiment-specific features	87.20	81.34	83.12	77.10	78.45	76.98	
TASCI with only GAT module	88.15	82.50	84.05	78.45	79.15	77.85	
TASCI Baseline (with all components)	89.10	83.38	84.81	78.63	79.08	77.27	

The third variation examined the TASCI model with only the GAT module. This configuration provided insights into the contribution of the GAT component alone. The results have shown accuracy of 88.15% and a macro-F1 score of 82.50% on the Restaurant dataset. For the Laptop dataset, the model achieved an accuracy of 84.05% and a macro-F1 score of 78.45%, while on the Twitter dataset, it reached 79.15% accuracy and a macro-F1 score of 77.85%. The GAT module, which enhances the model’s ability to understand context and relationships between different entities in the text, proved to be highly effective. Although the performance with only the GAT module did not match that of the full TASCI model, it still demonstrated a substantial improvement in both accuracy and F1 scores. Finally, the baseline TASCI model, which incorporates all components intent analysis, sentiment specific features, and the GAT module achieved the highest performance across all datasets. For the Restaurant dataset, the accuracy was 89.10% with a macro-F1 score of 83.38%. On the Laptop dataset, the model achieved 84.81% accuracy and a macro-F1 score of 78.63%. For the Twitter dataset, the results were 79.08% accuracy and 77.27% macro-F1 score. These results confirm that the full TASCI model, with all components included, delivers the best performance. The integration of intent analysis, sentiment-specific features, and the GAT module provides a comprehensive approach to sentiment classification, leading to improved accuracy and more detailed classification across different sentiment categories. Overall, the ablation study highlights the significant contributions of each component within the TASCI model. The removal of intent analysis leads to a marked decline in performance, underscoring its critical role in capturing nuanced user sentiments. While sentiment-specific features also contribute to the model’s effectiveness, their absence results in a less pronounced drop in performance compared to the removal of intent analysis. The GAT module alone enhances the model’s contextual understanding and relationship modeling capabilities, although it does not reach the same level of performance as the full model. The baseline TASCI model, incorporating all components, demonstrates the most robust performance, validating the effectiveness of its comprehensive design. This study provides valuable insights into the importance of each component and the necessity of their combined application for optimal sentiment analysis results.

Results of attention mechanism

This section presents the results of the attention mechanism in TASCI, attention is used in two ways, firstly it is used to highlight aspects in the sentence. The transformer automatically determines tokens in the given sentences which can be used to exact a certain aspect. Secondly, it is used to integrate aspect-level value with intention analysis. The outcome of the attention matrix shown in Fig. 4 visualization provides a clear view of how attention is distributed across the sentence, emphasizing specific aspects. In the generated attention matrix, you will observe a concentration of higher attention scores around the words corresponding to the identified aspects, such as “battery life” and “camera quality.” This indicates that the model pays more attention to these critical phrases, reflecting their importance in the context of the sentence. The matrix shows that attention is not confined to exact aspect words but is spread over adjacent words due to the inherent nature of attention mechanisms in neural networks. This results in a more realistic distribution where attention values gradually decrease as they move away from the key aspects. The visualization allows you to see how the model allocates focus, providing insight into which parts of the sentence are considered more significant in identifying aspects. This approach helps in understanding the model’s behaviour and its emphasis on specific components of the text.

Figure 4 Attention matrix for aspect analysis.

Conclusion

This article introduces TASCI, a Transformer-Based Aspect-Level Sentiment Classification model designed to enhance sentiment prediction accuracy and robustness through the integration of intent analysis. Traditional sentiment analysis techniques often fall short in addressing the nuanced relationship between a speaker’s intent and the sentiment expressed towards specific aspects of an entity. TASCI bridges this gap by leveraging a self-attention mechanism for precise aspect extraction and employing a Transformer-based model to capture and interpret the speaker’s intent from preceding sentences. This dual approach enables TASCI to provide a more contextually accurate reflection of user opinions. Our evaluation of TASCI across three benchmark datasets—Restaurant, Laptop, and Twitter—demonstrates its superior performance compared to several state-of-the-art models. TASCI achieves an accuracy of 89.10% and a macro-F1 score of 83.38% on the Restaurant dataset, surpassing the state-of-the-art. On the Laptop dataset, it attains an accuracy of 84.81% and a macro-F1 score of 78.63%, again outperforming leading models. Even on the Twitter dataset, known for its informal language and varied expressions, TASCI delivers results with an accuracy of 79.08% and a macro-F1 score of 77.27%.

These results highlight TASCI’s capability to handle complex sentiment expressions and demonstrate its effectiveness across diverse domains. The improvement in sentiment classification accuracy and macro-F1 scores underscores the importance of incorporating intent analysis into sentiment analysis models. TASCI’s consistent performance across different datasets not only showcases its robustness but also emphasizes its potential to offer more nuanced and context-aware sentiment insights. This advancement paves the way for further research into the integration of intent understanding in natural language processing tasks, potentially leading to more sophisticated and accurate sentiment analysis tools in the future. By advancing the state-of-the-art in aspect-level sentiment classification, TASCI sets a new benchmark for the field, inviting further exploration into the interplay between intent and sentiment. Its success highlights the value of context-aware models and opens avenues for developing even more refined methods for sentiment analysis, particularly in complex and varied linguistic contexts. Future research could build on TASCI’s framework to explore additional applications and enhancements, further advancing the capabilities of sentiment analysis in natural language processing.

Additional Information and Declarations

Competing Interests

The authors declare that they have no competing interests.

Author Contributions

Hassan Nazeer Chaudhry conceived and designed the experiments, performed the experiments, performed the computation work, authored or reviewed drafts of the article, and approved the final draft.

Farzana Kulsoom analyzed the data, prepared figures and/or tables, and approved the final draft.

Zahid Ullah Khan performed the experiments, analyzed the data, performed the computation work, authored or reviewed drafts of the article, and approved the final draft.

Muhammad Aman performed the experiments, performed the computation work, authored or reviewed drafts of the article, and approved the final draft.

Sajid Ullah Khan performed the computation work, authored or reviewed drafts of the article, and approved the final draft.

Abdullah Albanyan performed the computation work, authored or reviewed drafts of the article, and approved the final draft.

Data Availability

The following information was supplied regarding data availability:

The data is available at Zenodo: chaudhry,. hassan. nazeer. (2025). SemEval 2014 Task 4 [Data set]. Zenodo. https://doi.org/10.5281/zenodo.14655773.

The third party data is available at Kaggle: https://www.kaggle.com/datasets/charitarth/semeval-2014-task-4-aspectbasedsentimentanalysis.

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
