# Peer review of "TASCI: transformers for aspect-based sentiment analysis with contextual intent integration"

_PeerJ Computer Science, doi:10.7717/peerj-cs.2760_

## Round 0.1 · original submission · Minor Revisions

Dear Authors,

Thank you for the submission. The reviewers’ comments are now available. It is not suggested that your article be published in its current format. We do, however, advise you to revise the paper in light of the reviewers’ comments and concerns before resubmitting it. The followings should also be addressed:

1. "Related Work" section should be corretly written. Many of the sentence formation are incorrect.
2. "Figure 2" is not used in the main text.
3. Minor grammar and writing style errors should be corrected. Please pay special attention for correct writing, adjusting, and formatting. "4.3 Abalation Study" should be corrected.
4. Many of the equations are part of the related sentences. Attention is needed for correct sentence formation.
5. Equations should be used with correct equation number. Please do not use “as follows”, “given as”, etc. Explanation of the equations should also be checked. All variables should be written in italic as in the equations. Their definitions and boundaries should be defined. Provide proper reference to the governing equations.
6. Some of the variables in the equations listed in the definitions need to be explained. Some mathematical notations are not rigorous enough to correctly understand the contents of the paper. Please recheck all the definition of variables and further clarify these equations.
7. The running environment including software and hardware should be provided.

Warm regards,

Reviewer 1 ·

Basic reporting

The manuscript is written in clear, professional English, with precise terminology aligning with the fields of sentiment analysis and machine learning. The introduction effectively provides context for sentiment analysis, aspect-based sentiment analysis (ABSA), and intent-based sentiment analysis (IBSA), though it could benefit from explicitly linking the identified research gap to the objectives of the proposed TASCI model. The figures and tables are relevant, well-labeled, and contribute significantly to the understanding of the model's performance, though figure resolution should be checked for publication standards. The manuscript adheres to a logical structure, flowing seamlessly from the introduction to the methodology, results, and conclusions. The raw data and supplementary materials, including datasets and source code, are made available, meeting data transparency requirements, though ensuring all supplementary details, such as hyperparameters and configurations, are included will enhance reproducibility. The results are self-contained and tied to the hypotheses, with metrics like Accuracy, Precision, Recall, and F1-Score clearly presented; however, definitions of these metrics should be added for broader accessibility. The ablation study is thorough, validating the contributions of various model components. Additionally, the manuscript includes sufficient explanations of technical concepts, such as self-attention and GRU mechanisms, making it approachable for readers across different expertise levels. Expanding the conclusion to outline future research directions, such as the broader applicability of TASCI in other NLP tasks, would further strengthen the manuscript. Overall, the study adheres to PeerJ standards and effectively communicates its findings.

Experimental design

The paper makes a significant contribution to the field of context-aware sentiment analysis. The research question is clearly defined and aims to fill the knowledge gap in the literature. The TASCI model provides an innovative approach that includes contextual intent integration, which is lacking in traditional sentiment analysis methods. The results obtained on the restaurant, laptop and Twitter datasets demonstrate the accuracy and effectiveness of the proposed model. The originality of the research and the proposed solution are of significant value in the fields of NLP and machine learning.

In the method section, the components of the model are explained in detail and clearly, which increases the reproducibility. The integration of modern techniques such as the self-attention mechanism, GRU-based intent analysis and transformer-based sentiment classification supports the robustness and validity of the method. The benchmark datasets and performance metrics used increase the reliability of the analysis. However, a clearer specification of ethical standards regarding the use of data could strengthen the integrity of the study.

The results clearly demonstrate the superiority of the proposed model. The accuracy and Macro-F1 scores of TASCI show that it outperforms other existing models. However, more analytical and visual details of the results can deepen the reader's understanding. The strengths of the model include filling a gap in the literature, a detailed explanation of the method, and its success on different datasets. However, discussion of the limitations of the model and more detailed consideration of some of the shortcomings can broaden the scope of the study.
Overall, the article provides an innovative and effective solution to an important problem in the field of NLP and, with the proposed TASCI model, goes beyond existing methods in the literature. However, ethical standards, visual support and more detailed discussion of limitations would further strengthen the study.

Validity of the findings

The impact and novelty of the research have not been explicitly assessed within the manuscript. While the proposed TASCI model presents a clear contribution to the field by integrating intent analysis with aspect-based sentiment analysis, further elaboration on how this approach significantly advances the state-of-the-art or addresses practical challenges in real-world applications could enhance the impact. Meaningful replication of the study is encouraged, as the rationale for the model and its integration with existing methodologies is well-stated. The benefit of TASCI to the literature is evident, particularly in its ability to outperform prior models on benchmark datasets; however, its broader implications could be articulated more thoroughly.

All underlying data have been provided and are robust, statistically sound, and well-controlled. The use of benchmark datasets (Restaurant, Laptop, and Twitter) ensures that the results are comparable and relevant to the field of natural language processing. Additionally, the inclusion of detailed performance metrics (Accuracy and Macro-F1) for each dataset enhances the reliability of the findings.

The conclusions are well-stated and appropriately linked to the original research question. They are limited to supporting results, effectively summarizing the contributions and the superiority of TASCI in aspect-based sentiment classification. The conclusions align with the evidence presented and do not overreach, ensuring a balanced and evidence-based interpretation of the findings.

Reviewer 2 ·

Basic reporting

Language and Clarity: The manuscript is written in clear and professional language.
The technical concepts, methodologies, and results are well-articulated. However, minor
grammatical errors in sections like the introduction and methodology could be revised for
greater clarity.
Background and Literature Review: The introduction and related work sections
provide comprehensive context and cite relevant literature, establishing a strong
foundation for the proposed work. Figures and Tables: Figures and tables are relevant and well-labeled, supporting the narrative effectively. The attention matrix visualization is particularly insightful.

Experimental design

● Research Question and Novelty: The manuscript addresses the critical issue of
integrating intent analysis into aspect-based sentiment analysis (ABSA), a meaningful
extension to existing methods. The problem is well-defined, and the proposed TASCI
framework offers innovative solutions.
● Methodology: The approach is thoroughly explained, with detailed descriptions of the
self-attention mechanism, GRU-based intention extraction, and Transformer-based
sentiment classification. This clarity ensures the methodology is replicable.

Validity of the findings

Results and Comparisons: TASCI demonstrates superior performance in accuracy and
macro-F1 scores across all datasets, surpassing benchmarks. The ablation study
strengthens the manuscript by highlighting the importance of each component.

Statistical Rigor: The results are statistically sound, and the paper includes meaningful
metrics such as precision, recall, and F1 scores for detailed performance analysis.

Interpretability: Attention mechanism results provide insights into how the model
captures relevant aspects and integrates intent.

Additional comments

Strengths
● Innovative Approach: The integration of ABSA and intent analysis using Transformers
and GRUs is novel and enhances the model’s contextual understanding.
● Detailed Presentation: The methodology and results are well-presented, with clear
visualizations and explanations.


Weaknesses and Suggestions for Improvement
● Comparative Analysis: Although TASCI outperforms other models, a deeper discussion
of why specific components (e.g., intent integration) contribute to performance
improvements would strengthen the argument.
● Error Analysis: Including a detailed error analysis to understand misclassifications or
limitations could provide additional insights for future research.
● Real-World Applications: Expanding on potential practical applications or challenges of
deploying TASCI in real-world scenarios could add value.

---

## Round 0.2 · accepted · Accept

Dear Authors,

Thank you for clearly addressing the reviewers' comments. Your manuscript now seems sufficiently improved and ready for publication.

Best wishes,

Reviewer 1 ·

Basic reporting

Paper language has been improved. attention has been paid to the literature while making the desired changes. Tables, figures and materials presented in the whole study are adequate. They reflect the proposed methodology well. I think that the results will contribute to the academic literature.

Experimental design

With the revision, the study has academic competence. It is sufficient in terms of academic language, proposed method, method

Validity of the findings

With the revision, the study has academic competence

Additional comments

I think the study is publishable